# Falsification before Extrapolation in Causal Effect Estimation

**Zeshan Hussain**[*]
MIT CSAIL & IMES
Cambridge, MA
zeshanmh@mit.edu

**Michael Oberst**[*]
MIT CSAIL & IMES
Cambridge, MA
moberst@mit.edu

**Ming-Chieh Shih**[*]
National Dong Hwa University
Hualien, Taiwan
mcshih@gms.ndhu.edu.tw

**David Sontag**
MIT CSAIL & IMES
Cambridge, MA
dsontag@csail.mit.edu

## Abstract

*Randomized Controlled Trials* (RCTs) represent a gold standard when developing policy guidelines. However, RCTs are often narrow, and lack data on broader populations of interest. Causal effects in these populations are often estimated using observational datasets, which may suffer from unobserved confounding and selection bias. Given a set of observational estimates (e.g. from multiple studies), we propose a meta-algorithm that attempts to reject observational estimates that are biased. We do so using *validation effects*, causal effects that can be inferred from both RCT and observational data. After rejecting estimators that do not pass this test, we generate conservative confidence intervals on the *extrapolated* causal effects for subgroups not observed in the RCT. Under the assumption that at least one observational estimator is asymptotically normal and consistent for both the validation and extrapolated effects, we provide guarantees on the coverage probability of the intervals output by our algorithm. To facilitate hypothesis testing in settings where causal effect transportation across datasets is necessary, we give conditions under which a doubly-robust estimator of group average treatment effects is asymptotically normal, even when flexible machine learning methods are used for estimation of nuisance parameters. We illustrate the properties of our approach on semi-synthetic and real world datasets, and show that it compares favorably to standard meta-analysis techniques.

## 1 Introduction

Policy guidelines often rely on conclusions from Randomized Controlled Trials (RCTs), whether considering treatment decisions in healthcare, classroom interventions in education, or social programs in economics [31, 13, 43]. In healthcare, when a target population has reasonable overlap with the inclusion criteria of RCTs, current clinical treatment guidelines rely primarily on RCTs [23, 22]. For target populations not well-represented in RCTs, observational studies are often used to infer treatment effects. However, different observational estimates can give conflicting conclusions. We give an example of this tension when looking at a new chemotherapy for multiple myeloma.

**Example 1.1** (*Carfilzomib-based Combination Therapy for Newly Diagnosed Multiple Myeloma (NDMM)*)**.** Until 2020, the effect of Carfilzomib-based combination therapy in the NDMM subpopu-

---

[*]Equal contribution. Order determined alphabetically. For code and instructions on data access, please visit: https://github.com/clinicalml/rct-obs-extrapolation

36th Conference on Neural Information Processing Systems (NeurIPS 2022).

lation had not been studied via an RCT. However, a trial (ASPIRE) in 2015 measured the effect of Carfilzomib-based therapy on survival in Relapsed & Refractory Multiple Myeloma (RRMM) patients [58]. The CoMMpass trial, an observational dataset, was also available in which the Carfilzomib regimen was given to both NDMM and RRMM patients [36]. Several analyses on the CoMMpass dataset to estimate the effect of Carfilzomib-based therapy on NDMM patients led to different, sometimes opposing, conclusions on the benefit of the therapy in this subpopulation [33, 32].

A traditional meta-analysis approach would combine observational estimates under the assumption that differences arise only due to random variation, and not e.g., differences in confounding bias [24, Section 10.10.4.1]. This is unlikely to be true in practice. For instance, in Example 1.1, the two studies in question made different choices in e.g., how to adjust for confounders. *In this paper, we relax the assumption that all observational estimates are valid.* Instead, we assume that at least one observational estimate is valid across all subpopulations. In the context of Example 1.1, we might assume that at least one of the candidate observational studies yields consistent and asymptotically normal estimates of the effects in both the NDMM and RRMM populations. While we cannot *verify* that any given estimator is valid for all subpopulations, we can *falsify* this claim of validity if an estimator is inconsistent for the causal effects identified by the RCT (e.g., RRMM). Hence, we use the term *validation effects* to refer to causal effects in subpopulations that overlap between the observational and randomized datasets (e.g., RRMM), and use the term *extrapolated effects* to refer to those only covered by observational datasets (e.g., NDMM).

We propose a meta-algorithm that combines two key ideas: falsification of estimators, and pessimistic combination of confidence intervals. We first aim to falsify candidate estimators using hypothesis testing, rejecting those that fail to replicate the RCT estimates of validation effects. In Section 2.2, we motivate this approach with examples of observational estimates based on different causal assumptions, showing that hypothesis tests based on asymptotic normality can be applied even when causal assumptions fail to hold. Then, we combine accepted estimators to get confidence intervals on the extrapolated effects. Since failure to reject does not imply validity,[2] we return an interval that contains every confidence interval of the accepted estimators. We demonstrate theoretically that if *at least one* candidate estimator is consistent for both the validation and extrapolated effects, then the intervals returned by our algorithm provide valid asymptotic coverage of the true effects.

In scenarios where the covariate distribution differs across datasets, estimators that "transport" the causal effect should be used [42, 14, 15]. Furthermore, in the case of high-dimensional covariates, flexible machine learning methods are required to estimate nuisance functions, which can affect the hypothesis tests due to their slower convergence rates. In light of this, we adapt estimators of the average treatment effect in this setting to provide estimates of group-wise treatment effects, and show (via the framework of double machine learning [12, 55]) that this estimator enjoys asymptotic normality under mild conditions on convergence rates of the nuisance function estimators. Our conclusions are supported by semi-synthetic experiments, based on the IHDP dataset, as well as real-world experiments, based on clinical trial and observational data from the Women's Health Initiative (WHI), that demonstrate various characteristics of our meta-algorithm.

## 2 Setup and Motivating Examples

### 2.1 Notation and Assumptions

Let $Y \in \mathcal{Y}$ denote an outcome of interest, and $A \in \{0, 1\}$ denote a binary treatment. We use $Y_a$ to denote the potential outcome of an individual under treatment $A = a$. We use $X \in \mathcal{X}$ to denote all other covariates. To distinguish between different sampling distributions (i.e., datasets), we use the random variable $D \in \{0, \dots J\}$, where $J \geq 1$ is the number of observational datasets, and $D = 0$ is reserved for the sampling distribution of the randomized trial. We let $\mathbb{P}(Y_1, Y_0, Y, A, X, D)$ denote the joint distribution over all variables, including unobserved potential outcomes. For instance, $\mathbb{P}(Y_1, Y_0, X \mid D = 0)$ denotes the distribution of potential outcomes and covariates in the RCT.

We seek to estimate conditional average treatment effects for a finite set of $I$ subgroups $\{\mathcal{G}_i\}_{i=1}^I$. We assume subgroups are defined a-priori by a function $G : \mathcal{X} \mapsto \{1, \dots, I\}$, such that $G = i$ indicates that $X \in \mathcal{G}_i$. We use observational data precisely because not all groups are supported on the RCT

---

[2]For instance, we could fail to reject due to low power, or because falsification is impossible, due to differences in causal structure across subpopulations, as discussed in Appendix A.

dataset. To this end, we use $\mathcal{I}_R = \{i : \mathbb{P}(G = i \mid D = 0) > 0\}$ to denote the set of subgroups supported on the RCT dataset, and we let $\mathcal{I}_O$ denote the complement $\{1, \ldots, I\} \setminus \mathcal{I}_R$. We use $|\mathcal{I}_R|$ to denote the cardinality of a set, and assume that every observational dataset has support for all groups.

**Assumption 2.1** (Support). We assume that $\mathbb{P}(G = i, D = j) > 0$ for all $i \in \{1, \ldots, I\}$ and $j \in \{1, \ldots, J\}$, i.e., all observational datasets ($D \geq 1$) have support for all groups.

**Definition 2.1** (Validation and Extrapolated Effects). We define the group average treatment effect (GATE)[3] as

$$\tau_i := \begin{cases} \mathbb{E}[Y_1 - Y_0 \mid G = i, D = 0], & \text{if } i \in \mathcal{I}_R \\ \mathbb{E}[Y_1 - Y_0 \mid G = i, D = 1], & \text{if } i \in \mathcal{I}_O \end{cases} \tag{1}$$

and refer to $\tau_i$ for $i \in \mathcal{I}_R$ as a validation effect, and $\tau_i$ for $i \in \mathcal{I}_O$ as an extrapolated effect.

Here, we focus on discrete subgroups, in part to reflect the practical reality of comparing RCTs to observational studies, where we may have large observational datasets with rich covariates but only have access to the published results of the RCT, which often provides estimates (with confidence intervals) for subgroup effects but not the raw data itself [56, Figure 4, for example]. In Def. 2.1, we allow for the fact that different datasets may have different distributions of effect modifiers. To have a well-defined effect of interest, we have chosen the reference dataset $D = 1$ arbitrarily, but in principle we could choose any of the observational datasets. We discuss further nuances of this definition under Assumption 2.3. By Def. 2.1, we often write these effects as a vector $\tau \in \mathbb{R}^I$. We use $\hat{\tau}(k) \in \mathbb{R}^I$ to denote an estimator, where $k \in \{0, \ldots, K\}$, with $\hat{\tau}(0)$ reserved to denote the estimator derived from the RCT data. The remainder are observational estimators.[4] In general, we use "hat" notation to refer to estimators, and refer to their population quantities without a hat. We use $N_k$ to denote the number of samples used by each estimator. Throughout, we will assume that the RCT estimator is consistent.

**Assumption 2.2.** The RCT estimator $\hat{\tau}(0)$ is a consistent estimator of the (supported) dimensions of $\tau$, such that for each $i \in \mathcal{I}_R$, $\hat{\tau}_i(0)$ is consistent for $\tau_i$.

Below, our central assumption states that at least one observational estimator also enjoys consistency. We discuss examples of specific observational estimators in Section 2.2.

**Assumption 2.3.** There exists at least one observational estimator $\hat{\tau}(k) \in \mathbb{R}^I$, $k \geq 1$ that is a consistent estimator of $\tau \in \mathbb{R}^I$, such that for each $i \in \{1, \ldots, I\}$, $\hat{\tau}_i(k)$ is consistent for $\tau_i$.

*Remark* 2.1. Assumption 2.3 is our primary non-trivial assumption, and in Appendix B, we give one example of causal assumptions (for a given observational study) under which the entire GATE vector $\tau$ is **identifiable** from observational data, and give an **estimator** of the resulting observational quantity which is asymptotically normal [41, 42, 40, 15, 16]. In order to compare observational estimates with experimental ones, Assumption 2.3 requires not only that the observational data is free of confounding, but also that the causal effect can be transported to the RCT population. This can be done so long as relevant effect modifiers are observed in both the RCT and observational study, but the latter requirement is satisfied automatically (without requiring RCT data) if e.g., treatment effects are constant within each subgroup $G$, or if the distribution of effect modifiers is the same between the RCT and observational study, in which case $\mathbb{E}[Y_1 - Y_0 \mid D, G] = \mathbb{E}[Y_1 - Y_0 \mid G]$. This represents one (conservative) failure mode of our approach, in which we may reject an observational estimator due to failures in transportability, even if it yields unbiased estimates of the extrapolated effects.

Assumptions 2.2 and 2.3 imply that there exists an observational estimator $\hat{\tau}(k)$ such that both $\hat{\tau}_i(k)$ and the RCT estimate $\hat{\tau}_i(0)$ are both consistent for the validation effects $\tau_i, \forall i \in \mathcal{I}_R$. To validate this implication in finite samples, we will construct a statistical test to compare $\hat{\tau}_i(k)$ and $\hat{\tau}_i(0)$. Our general approach could be modified to use any valid test, but to facilitate further analysis, as well as explicit construction of confidence intervals, we additionally assume the following:

**Assumption 2.4.** All GATE estimators are pointwise[5] asymptotically normally distributed.

$$\sqrt{N_k}(\hat{\tau}_i(k) - \tau_i(k))/\hat{\sigma}_i(k) \xrightarrow{d} \mathcal{N}(0, 1) \tag{2}$$

for all $k \in \{0, , \ldots, K\}$, and for all $i$ in $\mathcal{I}_R$ if $k = 0$ (the RCT estimator), and otherwise for all $i \in \mathcal{I}_R \cup \mathcal{I}_O$. Here, $\xrightarrow{d}$ denotes convergence in distribution, and $\hat{\sigma}_i^2(k)$ is an estimate of the variance that converges in probability to $\sigma_i^2(k)$, the asymptotic variance of $\sqrt{N_k}(\hat{\tau}_i(k) - \tau_i(k))$.

---

[3] We use this term in line with the literature [11, 28, 38, 55] and to distinguish it from the CATE function.

[4] We define $\hat{\tau}(0)$ as a vector in $\mathbb{R}^I$ for simplicity of notation, allowing the entries $\hat{\tau}_i(0), i \in \mathcal{I}_O$ to be arbitrary.

[5] Here, "pointwise" refers to the fact that each subgroup effect estimate is asymptotically normal.

Assumption 2.4 requires each estimator $\hat{\tau}(k)$ to be consistent and asymptotically normal for some $\tau(k)$, which may **not** be equal to $\tau$. This is not a particularly strong assumption, as we discuss below.

## 2.2 Asymptotic Normality of Biased Estimators

In this section, we give two simple examples to illustrate the principle that multiple estimators $\hat{\tau}(k)$ may be asymptotically normal, even if they are asymptotically biased (i.e., $\tau(k) \neq \tau$). In both cases, there is a distinction between the *statistical* assumptions required to obtain asymptotic normality, and the *causal* assumptions required for $\tau(k)$ to identify the causal effect $\tau$. For simplicity in both examples, we restrict to the setting of comparing one-dimensional estimates $\tau(k) \in \mathbb{R}$, which estimate the GATE, $\tau$, in a single group $G = 1$ covered by all datasets. The statistical claims here also extend to GATE estimation with multiple groups [55].

**Example 2.1** (Variation in confounding across datasets)**.** Suppose that there is one estimator of the GATE per observational dataset, and each estimator seeks to estimate the population quantity, $\tau(k) = \mathbb{E}[g_k(1, X_k) - g_k(0, X_k) \mid G = 1, D = k]$, where $X_k$ denotes the controls used in each study, and $g_k(A, X_k) := \mathbb{E}[Y \mid A, X_k, D = k]$ and $m_k(X_k) := \mathbb{P}(A = 1 \mid X_k, D = k)$. We assume that $\eta < m_k(x) < 1 - \eta$ for some $\eta > 0$ for all $x, k$. Note that $\tau(k)$ is only a *statistical* quantity: identifying this with the *causal* quantity (the GATE) requires additional assumptions like unconfoundedness, that $Y_a \perp\!\!\!\perp A \mid X_k$ for the given dataset $D$. This assumption may hold for some datasets, but not others, particularly if the set of observed confounders $X_k$ differs across datasets.

Regardless of the interpretation of $\tau(k)$, one can construct estimators of it that are consistent and asymptotically normal using flexible machine learning estimators.[6] One approach, given in Chernozhukov et al. [12], is to use double machine learning (DML), which employs cross-fitting to produce estimates $\hat{\tau}(k)$ based on the doubly-robust score [45], while using plug-in estimates $\hat{g}_k, \hat{m}_k$ based on machine learning models. This approach achieves asymptotic normality, $\sqrt{N_k}(\hat{\tau}(k) - \tau(k))/\hat{\sigma}^2(k) \xrightarrow{d} \mathcal{N}(0, 1)$, under regularity conditions that allow for flexible machine learning estimators that converge at slower than parametric rates, and where $\hat{\sigma}^2(k)$ converges in probability to the variance of the doubly robust score [See Theorem 5.1 of [12], for additional details]. These results hold whether or not $\tau(k) = \tau$, as discussed in Footnote 9 of Chernozhukov et al. [12]. For simplicity, we have focused on the case where $\mathbb{E}[Y_1 - Y_0 \mid G = 1]$ is constant across datasets. When this does not hold, certain conditions enable valid transportation of treatment effects across datasets [16] with the use of transported estimators [15] (see Appendix B for details).

**Example 2.2** (Selection of Adjustment Strategy)**.** Consider the two causal graphs given in Figure 1, and assume that all variables are binary. Each graph suggests a different identification strategy for the causal effect, $\mathbb{E}[Y \mid do(A = a), G = 1]$. In Figure 1a, this is identified by the (observational) quantity $\mathbb{E}[Y \mid A = a, G = 1]$, and in Figure 1b, by front-door adjustment [39] as $\sum_M P(M \mid a, G = 1) \sum_{A'} \mathbb{P}(Y \mid M, A', G = 1)\mathbb{P}(A' \mid G = 1)$.

These observational quantities will typically differ: the one that represents the true interventional effect depends on which graph reflects the true causal structure. However, in the case where all variables are discrete and low-dimensional, we can still construct asymptotically normal estimators for both observational quantities.[7] For more complex settings (e.g., requiring regularized ML models for estimating conditional distributions) asymptotic normality has been established under certain conditions for general graphs [7, 29]

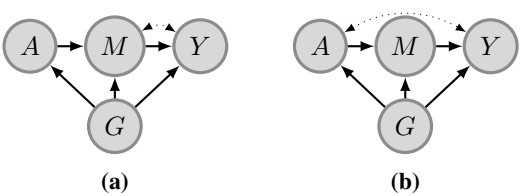

**Figure 1:** (Ex. 2.2) In (a), $M$ and $Y$ are confounded by unobservables (bi-directional dotted arrow). In (b), $A$ and $Y$ are confounded, but the causal effect is identified via front-door adjustment.

*Remark* 2.2. In each example, there are multiple estimators available, each asymptotically normal under basic statistical assumptions, but potentially biased in the sense that $\tau(k) \neq \tau$. In the first example, this bias occurs if $X$ is not sufficient to control for confounding in all observational

---

[6] A rich literature focuses on establishing such results, beyond the approach in this example [2, 20, 60, 37, 3].
[7] This follows from the use of maximum likelihood (i.e., empirical counts) for estimating each conditional distribution, and applying the delta method to the front-door estimator.

datasets. In the second, this bias arises in a given estimator if the causal graph is incorrectly specified. Assumption 2.3 corresponds to assuming that both the statistical assumptions and causal assumptions hold for one of the candidate estimators, e.g., $X$ is sufficient to control for confounding in at least one study (Example 2.1), or that one of the causal graphs is correct (Example 2.2).

## 2.3 Asymptotic Normality of GATE Estimators with Transportation

Example 2.1 assumes that $\mathbb{E}[Y_1 - Y_0 \mid G = i]$ is constant across datasets. In practice, it may be necessary to correct for differences (not captured by group indicators) between the observational and RCT populations. There exist estimators for the ATE in this setting under mild additional assumptions [15, 14]. These extend in a straightforward way to estimators of the GATE, but proving asymptotic normality is nuanced in high-dimensional settings when using flexible machine learning methods to estimate nuisance functions. For completeness, inspired by Semenova and Chernozhukov [55], we demonstrate that a doubly-robust GATE estimator for this setting is asymptotically normal under reasonable conditions (Assumption C.1 to C.5). Details on the estimator, and the corresponding proof of normality, are given in Appendix C, and may be of independent interest.

## 2.4 Testing for Bias under Asymptotic Normality

Under Assumption 2.4, each observational estimate $\hat{\tau}_i(k)$ can be compared to the estimate from the randomized trial $\hat{\tau}_i(0)$ for $i \in \mathcal{I}_R$, the groups with common support. Since the observational and randomized datasets are distinct, we can conclude that each $\hat{\tau}_i(k)$ is independent of $\hat{\tau}_i(0)$, and use this to test for the hypothesis that $\tau_i(k) = \tau_i$.

**Proposition 2.1.** *For an observational estimator $\hat{\tau}(k)$, assume Assumptions 2.2 and 2.4 hold. Furthermore, let $N = N_k + N_0$ with fixed proportions, where $N_k = \rho N, N_0 = (1 - \rho)N$ for $\rho \in (0, 1)$. Define the test statistic*

$$\hat{T}_N(k, i) := \frac{\hat{\tau}_i(k) - \hat{\tau}_i(0) - \mu_i(k)}{\hat{s}} \tag{3}$$

*where $\hat{s}^2 := \frac{\hat{\sigma}_i^2(k)}{N_k} + \frac{\hat{\sigma}_i^2(0)}{N_0}$ is the estimated variance, and $\mu_i(k) := \tau_i(k) - \tau_i$. This test statistic converges in distribution to a normal distribution as $N \to \infty$, $\hat{T}_N(k, i) \xrightarrow{d} \mathcal{N}(0, 1)$.*

We present the proof for Proposition 2.1 in Appendix D. This asymptotic normality allows for the construction of simple hypothesis tests. For instance, one can construct a Wald test for $H_0 : \tau_i(k) = \tau_i$, with asymptotic level $\alpha$ by setting $\mu_i(k) = 0$ in Equation (3) and rejecting $H_0$ whenever, $|\hat{T}_N(k, i)| > z_{\alpha/2}$, where $z_{\alpha/2}$ is the $1 - \alpha/2$ quantile of the normal CDF. Moreover, the asymptotic power of this test (the probability of correctly rejecting $H_0$) is given by

$$1 - \Phi\left(\frac{\mu_i(k)}{\sigma_{k,0}} + z_{\alpha/2}\right) + \Phi\left(\frac{\mu_i(k)}{\sigma_{k,0}} - z_{\alpha/2}\right) \tag{4}$$

where $\sigma_{k,0}^2 := \frac{\sigma^2(k)_i}{N_k} + \frac{\sigma_i^2(0)}{N_0}$ [see Theorems 10.4, 10.6 of [62]]. Likewise, Assumption 2.4 implies an asymptotic $1 - \alpha$ confidence interval for $\tau_i(k)$ as

$$[\hat{L}_i(k)(\alpha), \hat{U}_i(k)(\alpha)] := \left[\hat{\tau}_i(k) - \frac{z_{\alpha/2} \cdot \hat{\sigma}_i(k)}{\sqrt{N_k}}, \hat{\tau}_i(k) + \frac{z_{\alpha/2} \cdot \hat{\sigma}_i(k)}{\sqrt{N_k}}\right] \tag{5}$$

## 3 Meta-Algorithm for Conservative Extrapolation

In this section, we more formally introduce our algorithm (Algorithm 1). There are two primary steps: falsification of estimators, and combination of confidence intervals. First, we attempt to falsify candidate estimators via hypothesis testing, rejecting estimator $k$ whenever we are able to reject the null hypothesis $H_0 : \tau_i(k) = \tau_i, \forall i \in \mathcal{I}_R$. We use Bonferroni correction to control the false positive rate of the test. For the combination of confidence intervals, while we are unlikely to reject the "correct" estimator if one exists (Assumption 2.3), we may be unable to reject all "incorrect" (i.e., biased) estimators. This motivates the combination of confidence intervals (for the extrapolated effects) of the accepted estimators by taking the maximum and minimum bounds over all such intervals. Our main result characterizes the properties of our procedure, with proof in Appendix D.

**Algorithm 1** Extrapolated Pessimistic Confidence Sets

---

**Input:** Desired coverage $1 - \alpha$. For each $i \in \mathcal{I}_R$, RCT estimate $\hat{\tau}_i(0)$ and variance $\hat{\sigma}_i^2(0)$. For each $i \in \mathcal{I}_R \cup \mathcal{I}_O$, $K$ candidate estimators $\hat{\tau}_i(k)$ and variances $\hat{\sigma}_i^2(k)$. Sample sizes $N_0, \dots, N_K$.

**Initialize:** Empty candidate set $\hat{\mathcal{C}} \leftarrow \varnothing$

**for** $k = 1$ **to** $K$ **do**

    Compute $\hat{T}_N(k, i), \forall i \in \mathcal{I}_R$, with $\mu_i(k) = 0$ (Eq. 3)

    **if** $\forall i \in \mathcal{I}_R$, $\left| \hat{T}_N(k, i) \right| \leq z_{\alpha/4|\mathcal{I}_R|}$, **then** $\hat{\mathcal{C}} \leftarrow \hat{\mathcal{C}} \cup \{k\}$

**end for**

**for** $i \in \mathcal{I}_O$ **do**

    $\hat{L}_i \leftarrow \min_{k \in \hat{\mathcal{C}}} \hat{L}_i(k)(\alpha/2)$ and $\hat{U}_i \leftarrow \max_{k \in \hat{\mathcal{C}}} \hat{U}_i(k)(\alpha/2)$ (Eq. 5)

**end for**

**Return:** $\hat{L}_i, \hat{U}_i$ for each $i \in \mathcal{I}_O$.

---

**Theorem 3.1** (Properties of Algorithm 1). *Under Assumptions 2.1 and 2.2, the output of Algorithm 1 has the following asymptotic properties as $N \to \infty$, where $N$ denotes the total sample size, and the samples used for all estimators are of the same order $N_k = \rho_k N_0, \forall k \geq 1$, for some $\rho_k > 0$.*

1. *Under Assumptions 2.3 and 2.4, for each $i \in \mathcal{I}_O$,*

$$\lim_{N \to \infty} \mathbb{P}(\tau_i \in [\hat{L}_i, \hat{U}_i]) \geq 1 - \alpha \tag{6}$$

2. *Under Assumption 2.4, for each estimator where $\tau_i(k) \neq \tau_i$ for some $i \in \mathcal{I}_R$,*

$$\lim_{N \to \infty} \mathbb{P}(k \in \hat{\mathcal{C}}) = 0 \tag{7}$$

The first point says that for each extrapolated effect $\tau_i$, the coverage of the final confidence interval $[\hat{L}_i, \hat{U}_i]$ is at least $1 - \alpha$ in the limit. It follows from Assumption 2.3 and 2.4 that at least one estimator provides intervals $[\hat{L}_i(k)(\alpha/2), \hat{U}_i(k)(\alpha/2)]$ that achieve asymptotic coverage of $1 - \alpha/2$. The result follows from our choice of threshold for the significance test as well as application of union bounds. The second point says that we will reject estimators that are not consistent for the validation effects, in the limit. Assumption 2.4 ensures that Proposition 2.1 holds for all estimators, so that this rejection is a consequence of the asymptotic power in Equation (4), going to 1 for a fixed bias as $N \to \infty$.

*Remark* 3.1. Equations (4) and (5) are useful for building further intuition. All of the candidate confidence intervals shrink at a rate of $O(1/\sqrt{N})$ as the overall sample size increases. For sufficiently large $N$, the width of our generated intervals will depend largely on our power to reject biased estimators, which will be higher for observational estimates with larger biases for validation effects.

## 4 Semi-Synthetic Experiments

### 4.1 Setup of Simulation

We generate semi-synthetic RCTs and observational datasets with covariates from the Infant Health and Development Program (IHDP), a randomized experiment on premature infants assessing the effect of home visits from a trained provider on the future cognitive performance [8]. The outcomes are simulated. Our data generation is based on the partial IHDP dataset used in [25], which includes $n_0 = 985$ observation, 28 covariates, and a binary treatment variable. We construct a scenario where there are four subgroups, defined by the infant's birth weight and maternal marital status: (high [$\geq$ 2000g], married), (low [$< 2000$g], married), (high, single) and (low, single), which we shorthand as HM, LM, HS and LS. We include all subgroups in the observational studies, but exclude the latter two subgroups for the simulated RCT (i.e. only infants with married mothers are in the RCT).

For each simulated dataset, we generate 1 RCT and $K$ observational studies. For the observational studies, we resample the rows of the IHDP dataset to the desired sample size $n = r \cdot n_0$. We performed weighted sampling to induce a different covariate distribution for observational studies, such that male infants, infants whose mothers smoked, and infants whose mothers worked during pregnancy are less prevalent. Then, we introduce confounding in the observational data, generating $m_c$ continuous confounders and $m_b$ binary confounders. Finally, we simulate outcomes in each

dataset, modifying the response surface given in Hill [25]. In our experiments, we may choose to conceal some confounders in each observational study to mimic unobserved confounding, denoting the number of concealed variables across the $K$ studies as $\mathbf{c_z} = (c_{z1}, c_{z2}, ..., c_{zK})$. For further details on confounder generation, outcome simulation, and confounder concealment, see Appendix F. Data generation parameters include $K$, $r$, $m_c$, $m_b$, $\mathbf{c_z}$, and the significance level $\alpha$. By default, we set $K = 5$, $r = 10$, $m_c = 4$, $m_b = 3$, $\mathbf{c_z} = (0, 0, 2, 4, 6)$, and $\alpha = 0.05$. The full hyperparameter search is provided in Appendix F, and details of hyperparameter tuning can be found in Appendix C.

## 4.2 Implementation and Evaluation of Meta-Algorithm

To implement Algorithm 1, we first obtain GATE estimates for the four subgroups and their estimated variances in each observational study, combining techniques from the DML and trasportability literature [55, 15]. Estimation details are shown in Appendices B and C. For the RCT, we stratify the data into the subgroups HM, LM and estimate the GATEs as the difference of mean outcomes between the treated and untreated. The $z$ tests in Algorithm 1 are applied to both GATE estimates in the HM and LM subgroups ($|\mathcal{I}_R| = 2$), and the significance level of the tests is set at $\alpha/4$.

We evaluate performance using two main metrics: (1) the coverage probability of the output confidence intervals (ideally at least $1 - \alpha$), and (2) the width of the confidence intervals (narrower is better). In addition to assessing the intervals produced by Algorithm 1, which we call *Extrapolated Pessimistic Confidence Sets (ExPCS)*, we will evaluate intervals produced by a variant of our algorithm, called *Extrapolated Optimistic Confidence Sets (ExOCS)*. In *ExOCS*, after falsifying estimators, we combine confidence intervals using a random-effects meta-analysis on the non-falsified observational studies. We compare *ExPCS* and *ExOCS* against two baselines. *Meta-Analysis* is a random-effects meta-analysis on all observational studies, as described in Section 6, with heterogeneity variance estimated via the DerSimonian-Laird moment method [17]. This baseline is the current standard for aggregating observational study results. The second baseline, *Simple Union*, uses the maximum upper bound and minimum lower bound of the $1 - \alpha$ confidence intervals across all observational studies, with no falsification procedure.[8]

## 4.3 Results

We perform three semi-synthetic experiments to assess the performance of our proposed meta-algorithm under different scenarios. The first experiment applies our algorithm under the default settings given in Section 4.1. In the second experiment, we vary the sample size ratio between the observational studies and the original RCT, $r$, from 1 to 10. In the third experiment, we vary the proportion of biased observational studies by setting $\mathbf{c_z}$ to be $(0, 0, 0, 0, 0)$, $(0, 0, 0, 0, 3)$, $(0, 0, 0, 3, 3)$ or $(0, 3, 3, 3, 3)$, corresponding to $0, 1, 2, 4$ studies being biased out of a total of 5 observational studies. Results for the latter two experiments are shown over 100 simulations of the datasets. Results for all experiments are shown in Figures 2, 3, and Figure 5 in Appendix G, respectively. We observe the following:

*Meta-algorithm produces confidence intervals that cover the true GATE with nominal probability*: We demonstrate in Figure 2 the application of our meta-algorithm (*ExPCS*), a variant of it (*ExOCS*), and two other baselines on one dataset. Our goal is to produce narrow confidence intervals that still cover the true GATEs in the extrapolated subgroups. The confidence intervals of *ExPCS* cover the true GATEs in the extrapolated subgroups with reasonable widths. In contrast, intervals produced by *Meta-Analysis* fail to cover the true GATE in both extrapolated subgroups due to the false assumption of unbiasedness across all studies. The *ExOCS* approach produces narrow intervals for the extrapolated effects, though it barely covers the true effect in the HS subgroup. This hints at the need for a conservative combination of non-falsified studies. However, an overly conservative approach (e.g. *Simple Union*) produces wide intervals that may be of little use for meaningful inference.

Although *Meta-Analysis* produces confidence intervals with inadequate coverage, its intervals for the married subgroups still have considerable overlap with the intervals produced by the RCT. This suggests that testing the meta-analyzed GATE estimates against the RCT GATE estimates may not be enough to demonstrate their validity. Compared to our *ExPCS* intervals, the lower bounds of the

---

[8]Note that *Simple Union* combines $1 - \alpha$ confidence intervals, while our approach combines $1 - \alpha/2$ confidence intervals to account for the probability of rejecting the "correct" estimator, if one exists. As a result, *Simple Union* intervals do not always strictly cover the intervals produced by *ExPCS*.

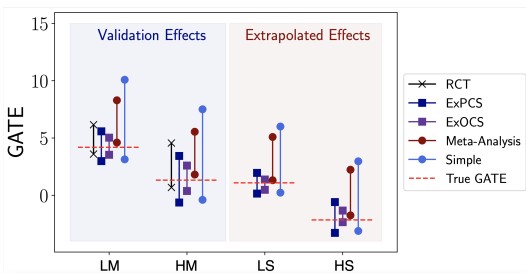

**Figure 2:** The confidence intervals for group average treatment effects (GATE) within the four subgroups output by our algorithm (*ExPCS*), our algorithm variant (*ExOCS*), random-effects meta-analysis on all observational studies (*Meta-Analysis*), simple union bound on all observational studies (*Simple*), and RCT, for one dataset generated using the default parameter settings laid out in Section 4.1. *LM, HM, LS, HS* represent four subgroups defined in Section 4.1.

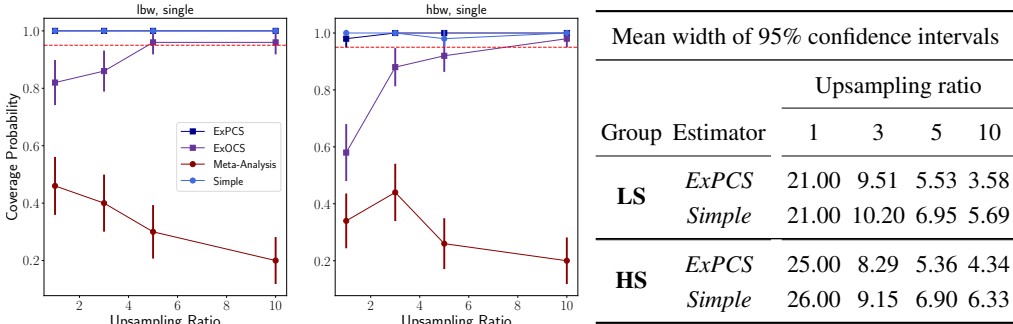

| | | Mean width of 95% confidence intervals | | | |
|---|---|---|---|---|---|
| | | Upsampling ratio | | | |
| Group | Estimator | 1 | 3 | 5 | 10 |
| **LS** | *ExPCS* | 21.00 | 9.51 | 5.53 | 3.58 |
| | *Simple* | 21.00 | 10.20 | 6.95 | 5.69 |
| **HS** | *ExPCS* | 25.00 | 8.29 | 5.36 | 4.34 |
| | *Simple* | 26.00 | 9.15 | 6.90 | 6.33 |

**Figure 3:** Coverage probabilities of confidence intervals shown as a function of the size of the observational studies relative to the RCT. Dotted red lines stand for 95% coverage. Vertical bars are the 95% confidence intervals of the coverage probabilities. LS / HS stand for groups with low / high birth weight and single mother. Between *ExPCS* and *Simple* which have adequate coverage, *ExPCS* generally has narrower intervals.

*Simple Union* intervals are higher in several subgroups, since we use a higher confidence level for the candidate intervals corresponding to each study to account for probable error in study falsification.

*An analysis of increasing observational study size*: In Figure 3, we find that the coverage of the *Meta-Analysis* intervals is quite low across all sample sizes and particularly decreases at higher sample sizes. This result is intuitive, as three out of five studies are biased, meaning that meta-analysis will converge to a biased estimate as the amount of data increases. One could attempt to fix this issue through *ExOCS*, which does meta-analysis after falsification. However, *ExOCS* has poor coverage when the sample size of the observational studies is small, since the falsification tests are underpowered (evidenced by the high probability of selecting biased studies in Appendix G, Table 2). Both *ExPCS* and the *Simple Union* intervals have adequate coverage across all sample sizes. However, the widths of the intervals reported at the bottom of Figure 3 show that *ExPCS* intervals are narrower when there is adequate power, i.e. at higher sample sizes. Ultimately, *ExPCS* will tend to provide intervals that cover the true effect regardless of sample size, and in the case we have sufficient power, these intervals will both have good coverage and narrower width, allowing for more meaningful inference.

## 5    Women's Health Initiative (WHI) Experiments

In order to assess our approach in a real-world setting, we use clinical trial and observational data available from the WHI. Each subgroup is supported in both RCT and observational data, which proves useful for evaluation. At a high level, we "hide" some number of subgroups from the RCT, estimate a confidence interval of the effect estimate using our algorithm on the remaining data, and compare the result to the hidden RCT estimate. We do this over a large set of possible "held-out" subgroups, yielding >2000 different scenarios on which to test our approach. Because the original observational datasets replicate the RCT results fairly well using standard methods, we create additional "biased" datasets by sub-selecting the original observational dataset in a way that induces

selection bias. We evaluate each method, for each held-out subgroup, according to the length of the intervals as well as coverage of the RCT point estimates. Below, we describe the specifics of the data, the experimental setup, and the main results of the analysis. For additional details on data preprocessing, setup, and evaluation, see Appendix E.

## 5.1 Setup

The Postmenopausal Hormone Therapy (PHT) trial, i.e. the RCT used in this analysis, was run on postmenopausal women aged 50-79 years who had an intact uterus. It studied the effect of hormone combination therapy on several types of cancers, cardiovascular events, and fractures. The observational study (OS) was run in parallel, had a similar follow-up time to the RCT, and tracked similar outcomes. In our analysis, we use a composite outcome, where $Y = 1$ if any of the following events are **observed** to occur in the first 7 years of follow-up, and $Y = 0$ otherwise: coronary heart disease, stroke, pulmonary embolism, endometrial cancer, colorectal cancer, hip fracture, and death due to other causes. This represents a binarization of the "global index" time-to-event outcome from the original study, where $Y = 0$ could also occur due to censoring. We establish treatment and control groups in the OS based on explicit confirmation or denial of usage of both estrogen and progesterone in the first three years. We use only covariates measured in both the RCT and OS to simplify analysis.

## 5.2 Evaluation

Our empirical evaluation consists of several steps. In the first step, we replicate the principal results from the PHT trial, given in Table 2 of [51], by fitting a doubly robust estimator (of the style given in Appendix C) on the WHI OS data. Then, while treating the WHI OS dataset as the "unbiased" observational dataset, we simulate additional "biased" observational datasets by inducing selection bias into the WHI OS. The exact mechanism of selection bias and its clinical intuition is given in Appendix E. Importantly, this is the only part of the evaluation that involves any simulation.

The second step is to construct a large suite of tasks on which to evaluate our method, by considering different sets of validation-extrapolation subgroups. To construct the subgroups, we consider all pairs of a selected set of binary covariates (see Appendix E.6), where each pair defines four subgroups. For example, one covariate pair is ("current smoker", "currently drinks alcohol"). We treat two of the subgroups as validation subgroups and two as extrapolated subgroups. For the latter groups, we apply our algorithm without access to the RCT data, and only use the RCT data for final evaluation. The total number of covariate pairs is 592, leading to 1184 distinct "tasks" (i.e., extrapolated groups). For each task, we evaluate ExPCS (our method), ExOCS, Simple, and Meta-Analysis (described in Section 4.2). Additionally, we evaluate an "oracle" method, which is identical to ExPCS, except that it always selects only the original observational study (i.e. the base WHI OS to which we have not added any selection bias). For each method, we compute the following metrics, averaged across all tasks – **Length**: length of the confidence interval, **Coverage**: percentage of tasks where the interval covers the RCT point estimate. In addition, we report the **Unbiased OS Percentage**: the percentage of tasks where the ExPCS approach retains the unbiased study after the falsification step.

## 5.3 Results

Table 1 reports the metrics above, averaged across all extrapolated subgroups.

*Compared to the "simple" baseline, our approach has better coverage with much shorter confidence intervals.* Our falsification procedure retains the unbiased observational study 99% of the time, yielding near-oracle coverage rates, but produces substantially shorter intervals than the "simple" baseline. Recall that the simple baseline takes a union over all $1 - \alpha$ intervals estimated from each observational dataset, while ExPCS takes a union of a smaller number of slightly wider $(1 - \alpha/2)$ confidence intervals.

|  | Coverage | Length | OS % |
|---|---|---|---|
| Simple | 0.39 | 0.416 | – |
| Meta-Analysis | 0.03 | 0.260 | – |
| ExOCS | 0.28 | 0.058 | – |
| **ExPCS (ours)** | 0.45 | 0.081 | 0.99 |
| Oracle | 0.44 | 0.068 | – |

**Table 1:** Coverage, length, and unbiased OS % of ExPCS and baselines. ExPCS achieves comparable coverage to the oracle method with highly efficient intervals. Additionally, we do not reject the unbiased OS in 99% of the tasks.

*Compared to the Meta-Analysis and ExOCS baselines, we achieve comparable (or much better) length with substantially better coverage.* In particular, compared to meta-analysis, we achieve tighter intervals and also cover the RCT estimate with higher frequency. This result is intuitive, since one will get a biased estimate if biased observational studies are included in the meta-analysis. Additionally, conservatively combining the non-falsified estimates (as opposed to *ExOCS*, which does a meta-analysis on the non-falsified estimates) is important to achieve good coverage (0.45 vs 0.28).

*We get comparable coverage and interval lengths to the oracle method.* Our coverage rate is nearly identical (0.45) to that of the oracle method (0.44), with intervals that are marginally wider (0.081 vs. 0.068). Our slightly improved coverage is possible due to the wider intervals. Note that our measure of "coverage" may be pessimistic, because we track coverage of the RCT point estimate, as opposed to the true causal effect (which is unknown), and the confidence intervals are designed to cover the latter. Indeed, we report the oracle method precisely as a means of providing a more suitable comparison. Overall, our real-world results suggest that our method of falsification followed by a conservative combination of intervals may be useful for biostatisticians and clinicians when doing meta-analyses.

## 6  Related Work

**Meta-analysis for combining observational estimates** Among the quantitative approaches for meta-analysis to account for potential bias, our *Meta-Analysis* baseline is standard for meta-analysis of observational data [24] to account for heterogeneity. Allowing for heterogeneity of treatment effects among studies produces wider confidence intervals and thus more conservative inference. If additional study-level covariates are available (e.g. study designs, drop-out rate), several approaches aim to adjust for potential bias, either by modeling the bias magnitude [19, 64, 1, 21], down-weighting studies with higher risk of bias [26, 35], or using Bayesian hierarchical regression to account for difference between subgroups of studies [44, 63]. Our work differs from these approaches, in that (1) we use information from outside the population of interest to assess bias, and (2) we do not place any assumptions on the patterns of bias across studies.

**Partial identification and sensitivity analysis** These methods seek to place bounds on causal effects when they cannot be point-identified. Our method can be seen as an alternative way of doing so, with a fundamentally different type of assumption. Methods for partial identification rely on having discrete variables and a known causal graph (typically including unobserved confounders) [18, Section 9]. Methods for sensitivity analysis, on the other hand, translate assumptions about the strength and nature of unobserved confounding into bounds on causal effects [47, 48, 65]. In contrast, we do not make any such assumptions, e.g., we allow for continuous variables, and when some candidate estimators are biased due to unmeasured confounding, we do not place any limit a-priori on the bias. An extended related work is given in Appendix H.

## 7  Discussion and Limitations

We have presented a meta-algorithm that constructs conservative confidence intervals for group average treatment effects of subgroups that are not represented in RCTs, but are represented in observational studies. Under the assumption that there exists at least one candidate estimator that is asymptotically normal and consistent for both the validation and extrapolated effects, these intervals will achieve the correct asymptotic coverage of the true effect. However, our method is not without limitations. Most notably, we may fail to reject the null hypothesis due to low power, e.g., when an observational estimate $\hat{\tau}(k)$ has high variance. In practice, we expect that our approach will be most useful when the observational studies in question have large sample sizes, leading to higher-precision estimates of potential bias, and smaller confidence intervals on the extrapolated effects. Our hope is that methods such as ours will lead to higher confidence in observational estimates when RCT data is available to falsify observational studies that do not replicate known causal relationships. Finally, great care should be taken to appropriately validate and soundly interpret the results of our method in practice, especially with more sensitive subgroups (e.g. with respect to race or gender).

**Acknowledgments**: We would like to thank Ahmed Alaa, Hunter Lang, Christina Ji, Hussein Mozannar, and other members of the Clincal Machine Learning group for helpful discussions and valuable feedback on the manuscript. ZH was supported by an ASPIRE award from The Mark Foundation for Cancer Research and by the National Cancer Institute of the National Institutes of Health under Award Number F30CA268631. The content is solely the responsibility of the authors and does not necessarily represent the official views of the National Institutes of Health. MO and DS were supported in part by Office of Naval Research Award No. N00014-21- 1-2807. MCS was supported by the LEAP program from the Ministry of Science and Technology in Taiwan.

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
