# OpenReview forum: "Falsification before Extrapolation in Causal Effect Estimation"
_NeurIPS.cc/2022/Conference — NeurIPS 2022 Accept_

### Official Review · Reviewer_hHvj · 2022-06-25

**Rating:** 6
**Confidence:** 3
**Soundness:** 3 good
**Presentation:** 3 good
**Contribution:** 3 good

**Summary:**

 Randomized controlled trials (RCTs) are considered the gold standard for studying the causal relationship between treatments and outcomes, and  Clinical Practice Guideline (CPG) policy recommendations are based on experimental results from RCTs. However, due to the cost (time and money) and ethical and methodological considerations, the populations in RCTs are narrow. Hence, the treatment effects outside of the support are missing. An alternative approach is to estimate treatment effects using observational data. However, the estimated treatment effects from these historical data might be biased due to the failure to control confounding effects or the existence of selection bias in the data. This work considers the problem of providing unbiased treatment effects, along with the confidence intervals, outside of the support of RCTs from the existing observational datasets. Given RCT data and multiple observational studies, this work first provides a hypothesis-testing technique to remove the observational estimates that are not consistent with the estimate of RCT. Under the assumption (Assumption 2.3) that there is a least one observational estimator that is a consistent estimator of RCT, i.e., the strong ignorability assumption holds, and the confidence intervals on the extrapolated treatment effects outside of the support of RCT is then can be provided. Throughout the experimental validation of the IHDP dataset, this work shows that their meta-algorithm can provide the confidence interval that covers the true GATE and with narrow width.

**Questions:**


This work provides a very solid meta-algorithm that is able to provide unbiased causal effects with confidence intervals from RCT and multiple observational datasets. However, the major question that comes to my mind is whether the use case of this approach is limited or not. I am not an expert in this area. Is it the case that many of the observational studies (or RCTs ) have a function mapping that $$ G: \mathcal{X} \mapsto \{1,\dots, I\}.$$ and the function mappings across studies are consistent so that the subsequent analysis can be provided from the meta-algorithm developed by the authors.

Perhaps it is challenging, but would it be possible to provide a real-world experimental validation using the publicly available clinical database, such as the MIMIV-IV database.


**Limitations:**

This work provides a meta-algorithm that can report unbiased causal effects outside of the support of RCT with confidence intervals. This work's only potential negative social impact might happen when they falsely accept the biased estimator from an observational study that is not consistent with the estimator of RCT. The author may provide more examples and cases of what would happen when this case is true.

**Strengths And Weaknesses:**

Strengths:

1. The paper is very written: Problem formulation, notations, assumptions, and derivations are clearly provided.
2. Empirical results are compared with existing meta-analysis and the comparison is comprehensive.

Weakness:

1. A lack of a real-world dataset is provided.

---

> ### Author Response · Authors · 2022-08-02
> **Response**
>
> Thank you for your helpful comments and feedback. Please see the main comment for empirical results on a real-world dataset and thoughts on the utility and practicality of our framing and setup.
>
> **This work provides a meta-algorithm that can report unbiased causal effects outside of the support of RCT with confidence intervals. This work's only potential negative social impact might happen when they falsely accept the biased estimator from an observational study that is not consistent with the estimator of RCT. The author may provide more examples and cases of what would happen when this case is true.**
>
> Thank you for this insightful comment. What you pose as a concern is only a problem if we **only** accept biased estimators. However, as long as we also do not reject the unbiased estimator, the resulting interval  will have the appropriate coverage of the true causal effect (since we take a union over all the non-rejected intervals). Indeed, in the above real-world experiment, we find that we include the unbiased observational study in >98% of the trials.

---

> > ### Comment · Reviewer_hHvj · 2022-08-08
> > **Thank you**
> >
> > I sincerely thank the authors for the answers. The additional real-world experiment is really helpful. I maintain my current score.

---

### Official Review · Reviewer_Gg3m · 2022-07-07

**Rating:** 6
**Confidence:** 4
**Soundness:** 3 good
**Presentation:** 3 good
**Contribution:** 2 fair

**Summary:**

Randomized controlled trials are high-standard with inclusion criteria in recruiting patients but may fail to include some heterogeneous patient subgroups in the full population. On the contrary, large-scale observational studies are likely to contain more diverse patient subgroups but they can be invalid to use due to hidden bias from some unmeasured confounders. The idea of this paper is to first validate the estimators from observational studies by comparing them with the estimator based on RCTs. This comparison is made on the patient subgroups that are observed in the RCTs. This step filters out the observational studies and their estimators that are inconsistent with the RCTs. After that, the authors use the non-rejected estimators and construct conservative confidence intervals to extrapolate the treatment effects for the subgroups that are not observed in the RCT.

**Questions:**

Why only consider group average treatment effects? could we extend the method to other causal parameters?

**Strengths And Weaknesses:**

Strengths
1. The paper recognizes the advantages and disadvantages of RCTs and observational studies and lets them complement each other in the method proposed to estimate group average treatment effects. The idea is interesting and well motivated!
2. I am convinced by Assumption 2.3 that at least one observational estimator is asymptotically normal and consistent for both the validation and extrapolated effects, and Assumption 2.2 which says that the RCT estimator is also consistent.  I think the assumptions are necessary so that at least in large samples, we can pick up the consistent observational estimator to estimate both effects.
3. The experiments uncover both the power and conservativeness of the proposed method. I still think the proposed idea is useful in practice.

Weaknesses
1. The groups $I_R$ and $I_O$ are given instead of being learnt from the data. I don't think this is often not the case in practice. We may know something about the effect heterogeneity but it is too strong to assume our knowledge is close to the ground truth.
2. The proposed method (Algorithm 1)  is not very novel and a bit trivial, which includes some standard techniques we often do in practice, e.g. constructing asymptotic normal estimators and doing a t-test. The author does not do much to make the method powerful, i.e., less conservative. I feel all the theoretical results are expected and follow from the DML literature.
3. The method looks quite conservative in Figures 3 and 4. The interval width is similar to the simple Union, which reports the union bounds of the confidence intervals of all observational studies, with no falsification procedure.

---

> ### Author Response · Authors · 2022-08-02
> **Response**
>
> Thank you for your feedback.  We address your first point in our general comment, and focus here on the remaining points.
>
> **The proposed method (Algorithm 1) is not very novel and a bit trivial, which includes some standard techniques we often do in practice, e.g. constructing asymptotic normal estimators and doing a t-test. The author does not do much to make the method powerful, i.e., less conservative. I feel all the theoretical results are expected and follow from the DML literature.**
>
> While we appreciate the perspective, we see the simplicity of the algorithm as a positive factor, not a negative one.  Likewise, from our perspective, using standard statistical tests (in the context of a broader algorithm) does not render the entire algorithm “not novel”, particularly given the lack of other algorithms that solve the same problem with similar guarantees.
>
> The main insight of Algorithm 1 lies not in the construction of new hypothesis tests (as you correctly point out, the tests we use are fairly standard), but rather the conservative strategy for combining intervals, which leverages the guarantees of the underlying tests.
>
> **The method looks quite conservative in Figures 3 and 4. The interval width is similar to the simple Union, which reports the union bounds of the confidence intervals of all observational studies, with no falsification procedure.**
>
> The difference is more striking in the real-data experiment (see our general comment, Table 1), where the length of the intervals output by ExPCS are ~20% the length of those output by the simple union.
>
> In general, this difference is problem-dependent, depending on the bias of individual estimators and the sample size: When it is easier to reject observational estimators, the intervals of ExPCS will tend to be shorter.  Regarding sample size, we demonstrate in Figure 3 (right) that ExPCS tends to have shorter intervals, relative to the simple union, as the sample size increases.  In general, ExPCS will also perform better in situations where some observational studies are more severely biased.

---

> > ### Comment · Reviewer_Gg3m · 2022-08-09
> > **Response**
> >
> > Thank you for your response to address my concerns. I adjusted the score accordingly.

---

### Official Review · Reviewer_cPqd · 2022-07-10

**Rating:** 6
**Confidence:** 3
**Soundness:** 3 good
**Presentation:** 2 fair
**Contribution:** 2 fair

**Summary:**

This paper proposes a meta-analysis for reliably extrapolating group level causal effects from multiple observational datasets when experimental data is available for some subgroups. The first part involves falsifying effect estimates from observational data that are biased. This is based on hypothesis testing where the statistic developed compares effect estimates from observational data to that of the RCT for groups that have experimental data. Under the assumption that at least one observational data exists for each group providing a consistent estimator (and assuming that all estimates are pointwise asymptotically normal), the statistic allows to reject biased estimates efficiently. Following this confidence intervals are generated using a simple algorithm that conservatively estimates the intervals based on the intervals of the observational data. Proposed method is evaluated on semi-synthetic IHDP data compared to simple meta-analysis and baselines that do not consist of falsification.

------------------------------------ Post rebuttal update ---------------------------------------------------------------------------------------------

I have read the full author response and I believe they address my major concerns with the paper. I have updated the score based on the response.

**Questions:**

1. Regarding my concern on practicality, can the authors suggest situations/RCTs that are explicitly designed to provide group level effects (this is different from subgroup analysis, which is not something one should do either way).

2. I would strongly encourage adding additional experiments, I have briefly seen the code and it will genuinely strengthen the paper.

3. Relating methods used for effect estimation in observational data to quality of falsification. This is clear to me based on the main assumption of asymptotic normality, it is however unclear if this is the only assumption needed? I am assuming all results use the DML estimators since they satisfy the assumption?

4. Conceptual clarification: Based on falsification, why can't a stronger statement be made about an estimator that aggregates the extrapolated effects?

5. Not sure why but some proofs in the appendix are made unnecessarily concise. Please expand and make everything clear.

**Limitations:**

I believe the authors have adequately described limitations of their work. Based on my clarification questions, please consider adding comments on practicality in the limitations section.

**Strengths And Weaknesses:**

Strengths:
1. The proposed falsification method is interesting although feels impractical due to concerns/clarifications I mention below.

2. The paper is well written, assumptions are clearly stated.

3. The simplicity of the approach is appealing.

4. Interesting experimental results.

Weaknesses:
1. It is really unclear how often RCT data could be available that could provide consistent group level effect estimates. This implies that the design of the RCT itself needs to be explicitly targeted for estimating group effects. Hence I am not sure how practical this approach really is.

2. Although the IHDP results are interesting, and evaluation specific to the data is thorough, I believe just one semi-synthetic evaluation is fairly limited to be convincing.

---

> ### Author Response · Authors · 2022-08-02
> **Response**
>
> Thank you for your feedback.  We address your questions (1) and (2) in our general comment, and focus here on addressing the remaining questions.
>
> **Relating methods used for effect estimation in observational data to quality of falsification. This is clear to me based on the main assumption of asymptotic normality, it is however unclear if this is the only assumption needed? I am assuming all results use the DML estimators since they satisfy the assumption?**
>
> *Regarding “unclear if this is the only assumption needed?”*:  The stated assumptions (2.1-2.4) are sufficient for the stated results (e.g., Theorem 3.1), as spelled out in the theorem statement.  If there is some other point of confusion regarding the necessity of assumptions, please feel free to raise this during discussion.
>
> *Regarding the use of DML estimators*: a range of estimators would also satisfy the asymptotic normality conditions, beyond DML-style estimators.  For instance, well-specified parametric models fit by maximum likelihood would also satisfy this condition, as discussed in Example 2.2 on lines 167-170.
>
> **Based on falsification, why can't a stronger statement be made about an estimator that aggregates the extrapolated effects?**
>
> A stronger statement on the performance of the ExPCS algorithm (e.g., probability of correctly rejecting all observational estimators that are not consistent for the true causal effect) would require additional assumptions regarding the power of each test, which depends on the level of bias and the variance of the estimators themselves.  However, such a result would not be difficult to derive: Equation (4) already provides the asymptotic probability of correctly rejecting an observational estimator.

---

> ### Author Response · Authors · 2022-08-08
> **Response #2**
>
> Thank you again for your review! One of your major concerns appeared to be the lack of a "real data" experiment, which we have now provided.
>
> Could you please let us know if this experiment (and our other clarifications below) has helped address your concerns? We'd be glad to provide any additional clarifications in the remaining time available for discussion.

---

### Official Review · Reviewer_icYd · 2022-07-12

**Rating:** 4
**Confidence:** 4
**Soundness:** 3 good
**Presentation:** 3 good
**Contribution:** 2 fair

**Summary:**

The authors propose an approach to estimating causal effect for populations for which only observational data exists, given that experimental data exists for a related population.

**Questions:**

Why is it more useful to describe to describe the proposed algorithm as a "meta-algorithm" rather than just a simple "algorithm"? Is the "meta" prefix intended to indicate that the procedure that you describe "is an algorithm about algorithms", that it is "an algorithm about meta-analysis", or something else?

**Limitations:**

To their credit, the authors are very clear about many of the limitations of their proposed approach. Clearly, the authors' approach requires that both experimental and observational data are available for a given (super) population of interest. The authors also assume that at least one observational estimate (among several) is "valid" across all subpopulations. This is a large assumption, given that any given estimate may have high bias or variance for any given subpopulation, and that those error properties are likely to vary substantially across different subpopulations. Finally, the authors assume that "every observational dataset has support for all groups." This seems unlikely. Furthermore, as the authors state clearly: "...we may reject an observational estimator due to failures in transportability, even if it yields unbiased estimates of the extrapolated effects."

Again, the authors are fairly clear about the limitations and assumptions of their proposed approach. This makes an extensive empirical evaluation all the more important as a demonstration that, even with these assumptions and limitations, the approach can produce accurate estimates of causal effect. Unfortunately, the empirical evaluation is limited to a single data set with known issues.

**Strengths And Weaknesses:**

**Strengths**

* The authors focus on how to exploit situations in which multiple data sets (both experimental and observational) are given, and increasingly realistic situation for problems of interest in social science, medicine, and other areas.

* The authors focus not just on obtaining point estimates, but on obtaining confidence intervals on those effects.

* The paper is well-grounded theoretically but also features an empirical demonstration of the approach on a well-known data set.

* The paper is clear about its assumptions.

**Weaknesses**

* The authors refer to observational estimates as "valid" or "invalid". Any estimate will have error due to bias and variance. More precise language would focus on the bias and variance of a given estimator, rather than a binary determination of "valid" or "invalid".

* The experiments in Section 4 are more of a demonstration than a convincing empirical evaluation of the proposed approach. The experiments employ only a single data set (IHDP), a practice which recently has been strongly critiqued (e.g., Curth et al. 2021). The approach used by the authors on IHDP could be applied to other RCT data (see Gentzel et al. 2021), but is not.  The result is that readers are left with little empirical evidence that the method works in practice, and theoretical treatment that makes a large set of assumptions that may (or may not) be valid.

* The homogeneity of causal effects is a standard (though highly suspect) assumption of most methods for estimating average treatment effect from observational studies. The authors authors assume that average treatment effect varies among subgroups, but that enough homogeneity exists that valid extrapolations can be made. However, RCTs and observational studies often differ in ways that go far beyond the randomization of treatment assignment. Randomized experiments often involve other artificial conditions that are not replicated in observational studies. Meanwhile observational studies are often substantially different from each other (and very different from experimental settings).

* For most of the paper, the authors oddly refrain from naming the proposed algorithm. This leads to odd linguistic constructions such as a section headers that read "Implementation and Evaluation of Meta-Algorithm" and "Meta-algorithm produces confidence intervals that cover the true GATE with nominal probability". Then, in section 4.2, the authors name the algorithm (Extrapolated Pessimistic Confidence Sets (ExPCS)). Use the name early and throughout the paper.

* The authors use of the term "meta-algorithm" seems unnecessary.

* The use of hypothesis tests, particularly given the large literature on the limitations of this basic framework, is open to criticism.

---

> ### Author Response · Authors · 2022-08-02
> **Response**
>
>
> We thank the reviewer for their helpful comments and feedback. We will respond to each point in turn.
>
> **The authors refer to observational estimates as "valid" or "invalid"... more precise language would focus on the bias and variance of a given estimator...**
>
> While we appreciate the feedback, we do give precise definitions in terms of asymptotic consistency, the property that the estimate converges to the true effect as the sample size becomes large.
>
> **The experiments in Section 4 are more of a demonstration than a convincing empirical evaluation of the proposed approach.**
>
> See our general comment about the real-data experiment.
>
> **The homogeneity of causal effects is a standard (though highly suspect) assumption of most methods for estimating average treatment effect from observational studies**
>
> To clarify, we are not claiming homogeneity of treatment effects within subgroups. Subgroups serve as a means of defining a reference covariate distribution, where individuals within that subgroup can still have heterogeneous treatment effects.
>
> **RCTs and observational studies often differ in ways that go far beyond the randomization of treatment assignment**
>
> The reviewer makes the point that RCTs and and observational studies differ in ways beyond randomization of treatment assignment, and RCTs often “involve other artificial conditions that are not replicated in observational studies.” We would like to point out that there is a wealth of literature on RCT emulation using observational studies, see [2-5] for examples. Furthermore, papers on trial emulation show that when construction of the observational cohort data respects the design of the RCTs, the effect estimates of the RCTs can actually be replicated [4,5]. In our own empirical results above on the WHI dataset, we are able to replicate the RCT results (i.e. the ATE estimates for multiple outcomes) reported in [1] after careful construction of the observational cohort to match the inclusion criteria of the RCT and aligning the OS with the RCT in terms of follow-up.
>
> **The authors’ use of the term "meta-algorithm" seems unnecessary.**
>
>
> We used the term originally to mean “an algorithm that takes other algorithms (i.e., estimators) as input”, where the particular estimators for GATE can be flexibly chosen, as  long as they satisfy Assumptions 2.3 and 2.4.  However, we can see why this might lead to some confusion, and will remove this language from the revision.
>
> **For most of the paper, the authors oddly refrain from naming the proposed algorithm.**
>
> Thank you for pointing this out - we will name the algorithm earlier in the revised version.
>
> **The use of hypothesis tests, particularly given the large literature on the limitations of this basic framework, is open to criticism.**
>
> While we appreciate the concern, hypothesis testing (and construction of confidence intervals based on asymptotics) is nonetheless widely used in medicine and other fields (e.g., A/B testing) where we see our method being applied.
>
>
> *[1] Rossouw, Jacques E., et al. "Risks and benefits of estrogen plus progestin in healthy postmenopausal women: principal results From the Women's Health Initiative randomized controlled trial." Jama 288.3 (2002): 321-333.*
>
> *[2] Franklin, Jessica M., et al. "Emulating randomized clinical trials with nonrandomized real-world evidence studies: first results from the RCT DUPLICATE initiative." Circulation 143.10 (2021): 1002-1013.*
>
> *[3] Hernán, Miguel A., et al. "Specifying a target trial prevents immortal time bias and other self-inflicted injuries in observational analyses." Journal of clinical epidemiology 79 (2016): 70-75.*
>
> *[4] García-Albéniz, Xabier, John Hsu, and Miguel A. Hernán. "The value of explicitly emulating a target trial when using real world evidence: an application to colorectal cancer screening." European journal of epidemiology 32.6 (2017): 495-500.*
>
> *[5] Dickerman, Barbra A., et al. "Avoidable flaws in observational analyses: an application to statins and cancer." Nature medicine 25.10 (2019): 1601-1606.*

---

> ### Author Response · Authors · 2022-08-08
> **Response #2**
>
> Thank you again for your review! One of your major concerns appeared to be the lack of a "real data" experiment, which we have now provided.
>
> Could you please let us know if this experiment (and our other clarifications below) has helped address your concerns?  We'd be glad to provide any additional clarifications in the remaining time available for discussion.

---

### Author Response · Authors · 2022-08-02
**Discussion + New "Real Data" Experiment**

We thank you for your constructive feedback on our manuscript. In this rebuttal, we will address two points that the majority of the reviewers have brought up as weaknesses of the work: **1] concerns with respect to the “practicality” of our setup, namely how often RCTs provide consistent, group-level effect estimates** and **2] limited empirical evaluation of our approach, particularly with a “real-world” dataset**. We will address these points as a general comment to all reviewers and then provide individualized responses in additional comments below.

# The practicality of our setup

In our setup, we assume that average treatment effect estimates for pre-specified subgroups $I_R$ are available for the RCT data.  Here, we do allow the number of subgroups in $I_R$ to be 1, so our meta-algorithm is still valid even if **only** the average treatment effect (ATE) estimate is available for the RCT. In the case where there is more than one subgroup, the group average treatment effect (GATE) estimates can come from a prespecified subgroup analysis on the RCT data, which is common for large-scale clinical trials (see e.g., [1-3]).  By using only pre-specified subgroup analyses and not post-hoc subgroup analyses, we avoid possible bias of the GATE estimates from selective reporting, which appeared to be a concern of reviewer cPqd.

We do ***not*** require that these subgroups capture all effect heterogeneity (e.g., that effects are homogenous within pre-specified subgroups), which appeared to be a concern of reviewer Gg3m. If they do, the chances of falsification by testing GATEs may be improved, but the meta-algorithm is valid under any subgroup stratification. Additionally, as reviewer hHvj has mentioned, the grouping G should align with the RCT and observational studies. Practically, the groupings for $G \in I_R$, are determined by existing subgroup analyses in the literature. In the case where we have individual data for the observational studies, it is straightforward to produce GATE estimates for $G \in I_R$ and construct $G \in I_O$ that is shared among the observational studies for extrapolation and combination.

*[1] Pavord, I. D., et al.  (2020). Predictive value of blood eosinophils and exhaled nitric oxide in adults with mild asthma: a prespecified subgroup analysis of an open-label, parallel-group, randomized controlled trial. The Lancet Respiratory Medicine, 8(7), 671-680.*

*[2] Oskarsson, P., et al.  (2018). Impact of flash glucose monitoring on hypoglycaemia in adults with type 1 diabetes managed with multiple daily injection therapy: a pre-specified subgroup analysis of the IMPACT randomized controlled trial. Diabetologia, 61(3), 539-550.*

*[3] Solomon, A et al. (2018). Effect of the apolipoprotein E genotype on cognitive change during a multidomain lifestyle intervention: a subgroup analysis of a randomized clinical trial. JAMA neurology, 75(4), 462-470.*

# Summary of new experimental results

In order to assess our approach on real clinical data, we use RCT + observational data available from the Women’s Health Initiative (WHI).  Here, each subpopulation is covered by both RCT and observational data, which is useful for evaluation; we can “hide” a subpopulation in the RCT, estimate a confidence interval using our algorithm applied to the the remaining RCT + observational data, and compare the result to the hidden RCT estimate.

We do this over a large set of possible “held-out” subgroups, yielding >2000 different scenarios on which to test our approach.  Because the original observational datasets replicate the RCT results fairly well using standard methods, we create additional “biased” datasets by sub-selecting the original observational dataset in a way that induces selection bias.  We evaluate each method, for each held-out subgroup, according to the length of the intervals as well as coverage of the RCT point estimates.

In the aggregate, we observe (see Table 1, copied below for convenience) that our approach:

* Has much shorter intervals (~20% as large) than the “Simple Union” approach, with comparable coverage of the RCT effect estimates.
* Has much shorter intervals (~30% as large) than the Meta baseline, and superior coverage of the RCT effect estimates.
* Has comparable intervals to the ExOCS baseline, but with superior coverage of the RCT effect estimates.
* Has comparable coverage and intervals compared to an oracle method that always selects only the estimator from the original observational study (excluding all estimators from the biased datasets) in the falsification step.

| Method       | Coverage | Length |
|--------------|----------|--------|
| *Oracle*     | 0.44     | 0.068  |
| ExPCS (ours) | 0.45     | 0.081  |
| ExOCS        | 0.28     | 0.058  |
| Meta         | 0.03     | 0.260  |
| Simple       | 0.39     | 0.416  |

If accepted, we will include these results (below) in the camera-ready version, where there is space for an additional page of content.

---

> ### Author Response · Authors · 2022-08-02
> **Details of Experiment (Part 1)**
>
> ## Details of WHI experiment
>
> We begin by addressing concerns with our empirical evaluation by assessing our algorithm on clinical trial data and observational data available from the Women’s Health Initiative (WHI). The RCTs were run by the WHI via 40 US clinical centers from 1993-2005 (1993-1998: enrollment + randomization; 2005: end of follow-up) on postmenopausal women aged 50-79 years, and the observational dataset was designed and run in parallel on a similar population. Note that this data is publicly available to researchers and requires only an application on BIOLINCC (https://biolincc.nhlbi.nih.gov/studies/whi_ctos/).
>
> ### Data
>
> WHI RCT — There are three clinical trials associated with the WHI. The RCT that we will be leveraging in this set of experiments is the Postmenopausal Hormone Therapy (PHT) trial, which was run on postmenopausal women aged 50-79 years who had an intact uterus. This trial included a total of $N_{HT} = 16608$ patients. The intervention of interest was a hormone combination therapy of estrogen and progesterone. Specifically,  post-randomization, the treatment group was given 2.5mg of medroxyprogesterone as well as 0.625mg of estrogen a day. The control group was given a placebo. Finally, there are several outcomes that were tracked and studied in the principal analysis done on this trial [1]. These outcomes are of three broad categories: a) cardiovascular events, including coronary heart disease, which served as a primary endpoint b) cancer (e.g. endometrial, breast, colorectal, etc.), and c) fractures.
>
> WHI OS — The observational study component of the WHI tracked the medical events and health habits of $N = 93676$ women. Recruitment for the study began in 1994 and participants were followed until 2005, i.e. a similar follow-up to the RCT. Follow-up was done in a similar fashion as in the RCT (i.e. patients would have annual visits, in addition to a “screening” visit, where they would be given survey forms to fill out to track any events/outcomes). Thus, the same outcomes, including cancers, fractures, and cardiovascular events, are tracked in the observational study.
>
> ### Outcome
>
> The outcome of interest in our analysis is a “global index”, which is a summary statistic of several outcomes, including coronary heart disease, stroke, pulmonary embolism, endometrial cancer, colorectal cancer, hip fracture, and death due to other causes. Events or outcomes are tracked for each patient, and are recorded as “day of event/outcome” in the data, where the initial time-point for follow-up is the same for both the RCT and OS. At a high level, the “global index” is essentially the minimum “event day” when considering all the previously mentioned events.
>
> We binarize the “global index,” by choosing a time point, $t$, before the end of follow-up and letting $Y=1$ if the observed event day is before $t$ and $Y=0$ otherwise. Thus, we are looking at whether the patient will experience the event within some particular period of time or not. We set $t = 7$ years. Note that we sidestep censorship of a patient before the threshold by defining the outcomes in the following way: $Y=1$ indicates that a patient is observed to have the event before the threshold, and $Y=0$ indicates that a patient is not observed to have the event before the threshold. We apply this binarization in the same way for both the RCT and OS. Extending our method to a survival analysis framing is beyond the scope of this paper, but an interesting direction for future work.
>
> ### Intervention
>
> Recall from above that the intervention studied in the RCT was 2.5mg of medroxyprogesterone + 0.625 mg of estrogen and the control was a placebo pill. The RCT was run as an “intention-to-treat” trial. To establish “treatment” and “control” groups in the OS, we leverage the annual survey data collected from patients and assign a patient to the treatment group if they confirm usage of both estrogen and progesterone in the first three years. A patient is assigned to the control group if they deny usage of both estrogen and progesterone in the first three years. We exclude a patient from the analysis if she confirms usage of one and not the other OR if the field in the survey is missing OR if they take some other hormone therapy. We end up with a total of $N_{obs} = 33511$ patients.
>
> ### Data Processing + Covariates
>
> We use only covariates that are measured both in the RCT and OS to simplify the analysis. Because this information is gathered via the same set of questionnaires, they each indicate the same type of covariate. In other words, there is consistency of meaning across the same covariates across the RCT and the OS. We end up with a total of 1576 covariates.

---

> > ### Author Response · Authors · 2022-08-02
> > **Details of Experiment (Part 2)**
> >
> > ### Experimental Setup
> >
> > In this analysis, we aim to show the effectiveness of our approach in a real-world setting compared to the baselines (i.e. Simple, Meta-analysis, ExOCS). We detail how we evaluate our method below.
> >
> > Our experimental workflow consists of the following steps:
> >
> > **Step 0**: Replicate the principal results from the PHT trial, given in Table 2 of [1], using the WHI OS data. In this step, we fit a doubly robust estimator of the style given in Appendix C of the supplement.
> >
> > **Step 1**: While treating the WHI OS dataset as the “unbiased” observational dataset (hence the need for Step 0), simulate additional “biased” observational datasets by inducing bias into the WHI OS. We construct four additional “biased” datasets (for a total of five observational datasets, including the WHI OS datasheet), where we use the following procedure to induce selection bias – of the people who were not exposed to the treatment and did not end up getting the event, we drop each person with some probability, $p$. We set $p = [0.1, 0.3, 0.5, 0.7]$ to get the four additional observational datasets.
> >
> > This type of selection bias may reflect the following clinical scenario: consider a patient who is relatively healthy who does not end up taking any hormone therapy. This patient might enroll initially in the OS, but may drop out or stop responding to the surveys. If the committee running the study does not explicitly account for this drop-out rate, then the resultant study will suffer from selection bias. [2] detail additional examples of selection bias that can occur in observational studies.
> >
> > Importantly, this part is the only part of our setup that involves any simulation. However, in order to properly evaluate our method, we would need to know which datasets are biased and unbiased in our set. Thus, we opt to simulate the bias.
> >
> > **Step 2**:  In this step, we wish to run our procedure over “multiple trials,” generating confidence intervals on the treatment effect for different subgroups. To do so, we compile a list of covariates, taking both from [3] as well as covariates with high feature importance in both the propensity score model and response surface model from the estimator in Step 0. We generate all pairs from this list and use each pair to generate four subgroups. We treat two of the subgroups as validation subgroups and two of them as extrapolated subgroups in that we “hide” the RCT data in those subgroups when fitting our doubly robust transported estimator. (This gives us the benefit of knowing the RCT result for the extrapolated subgroups, which is useful in evaluation). Pairs that don’t have enough support (threshold of 400 observations) in each group are removed. The total number of “trials” (or covariate pairs) that we have is 592 (and therefore 2368 subgroups).
> >
> > **Step 3**: For each of the covariate pairs, we will evaluate ExPCS (our method), ExOCS, Simple, and Meta. Additionally, we evaluate an “oracle” method, which always selects only the original observational study (i.e. the base WHI OS to which we have not added any selection bias) and reports the interval estimate computed on this study. To evaluate these methods, we will treat the RCT point estimates as “correct.” For each, we compute the following metrics:
> >
> > * **Length** – length of the confidence interval for the subgroup
> > * **Coverage** – percentage of trials for which the method’s interval covers the RCT point estimate
> >
> > Additionally, we report, across all trials, the percentage at which our approach retains the unbiased study after the falsification step (we call this metric: **Unbiased OBS Percentage**).
> >
> > Note that we utilize sample splitting when running the above procedure. Namely, we use 50% of the data as a “training” set, where we experiment with different classes of covariates and different types of bias, and then reserve 50% of the data as a “testing” set, on which we do the final run of the analysis and report results. All nuisance functions in the OBS doubly robust estimator are fit with a Gradient Boosting Classifier with significant regularization. In practice, we found that any highly-regularized tree-based model works well.
> >
> > *[1] Rossouw, Jacques E., et al. "Risks and benefits of estrogen plus progestin in healthy postmenopausal women: principal results From the Women's Health Initiative randomized controlled trial." Jama 288.3 (2002): 321-333.*
> >
> > *[2] Banack, Hailey R., et al. "Investigating and remediating selection bias in geriatrics research: the selection bias toolkit." Journal of the American Geriatrics Society 67.9 (2019): 1970-1976.*
> >
> > *[3] Schnatz, Peter F., et al. "Effects of calcium, vitamin D, and hormone therapy on cardiovascular disease risk factors in the Women's Health Initiative: a randomized controlled trial." Obstetrics and gynecology 129.1 (2017): 121.*

---

> > > ### Author Response · Authors · 2022-08-02
> > > **Detailed Results of Experiment**
> > >
> > >
> > > ### Results
> > >
> > > We report, in Table 1, the above metrics averaged across all extrapolated groups.
> > >
> > > | Method       | Coverage | Length | Unbiased OBS Percentage |
> > > |--------------|----------|--------|-------------------------|
> > > | *Oracle*     | 0.44     | 0.068  | -                       |
> > > | ExPCS (ours) | 0.45     | 0.081  | 0.988                   |
> > > | ExOCS        | 0.28     | 0.058  | -                       |
> > > | Meta         | 0.03     | 0.260  | -                       |
> > > | Simple       | 0.39     | 0.416  | -                       |
> > > **Table 1**
> > >
> > > We can glean a few high-level takeaways from these results:
> > >
> > > **Compared to the “simple” baseline, our approach has better coverage with much shorter confidence intervals.** Recall that the simple baseline takes a union over all intervals estimated from each observational dataset. Thus, this result indicates that our falsification procedure is important to get tighter intervals but still by and large retains the unbiased observational study, which is important to getting an unbiased estimate.
> > >
> > > **Compared to the Meta and ExOCS baselines, we get comparable (or much better in the case of Meta) length with substantially better coverage.** In particular, compared to meta-analysis, which is a standard procedure done in the biostatistics and epidemiology communities as discussed in the main paper’s Related Work, we get much tighter intervals and also cover the RCT estimate with higher frequency. This result is intuitive, since one will get a biased estimate if biased observational studies are included in the meta-analysis.
> > >
> > > **We get comparable coverage and interval lengths to the oracle method.**  Our coverage rate is nearly identical (0.45) to that of the oracle method (0.44), with intervals that are marginally wider (0.081 vs. 0.068).  Note that our slightly improved coverage is possible due to the wider intervals.
> > >
> > > Note that our measure of “coverage” may be pessimistic, because we track coverage of the RCT point estimate, as opposed to the true causal effect (which is unknown), and the confidence intervals are designed to cover the latter, not the former.
> > >
> > > Overall, we find that our real-world results are reassuring and suggest that our method of falsification followed by a combination of intervals may be useful in how biostatisticians and clinicians do meta-analyses of observational studies.

---

### Meta-Review · Area_Chair_AZmZ · 2022-08-26

**Recommendation:** Accept
**Confidence:** Certain

**Metareview:**

The authors propose an approach for estimating causal effects when both observational and limited experimental data exists. The authors propose falsifying effect estimates from observational data before using the effect estimate on other populations. This is an important idea that may improve reliability of causal inference. The authors provide confidence intervals for the proposed procedure. The considered problem is of clear importance; and the simplicity of its approach is appealing (cPQd). There have been some concerns about the limited empirical evaluation (icYd). The authors provide additional numerical evidence during the rebuttal period. This evidence should be added to the appendix for the camera-ready version.

Note: The reviewer most critical of the paper (rating 4, icYd) does not seem to have updated their score post-rebuttal.

**Award:**

No

---

### Decision · Program_Chairs · 2022-09-14

Accept